# DAPE cloning with modified primers for producing designated lengths of 3′ single-stranded ends in PCR products

**Seoee Lee**[1], **Hyunyoung Kim**[1], **Aqsa**[1], **Kwangjin Jeung**[2], **Minho Won**[2,3]*, **Hyunju Ro**[1]*

**1** Department of Biological Sciences, College of Bioscience and Biotechnology, Chungnam National University, Daejeon, Korea (ROK), **2** Biotechnology Process Engineering Center, KRIBB, Cheongju, Korea (ROK), **3** Department of Biochemistry, College of Natural Sciences, Chungnam National University, Daejeon, Korea (ROK)

* rohyunju@cnu.ac.kr (HR); minhowon@kribb.re.kr (MW)

**Data Availability Statement:** All relevant data are within the paper and its Supporting Information files.

## Abstract

For in vitro DNA assembly, enzymes with exonuclease activities have been utilized to generate relatively long recessed ends on DNA fragments, which can anneal to other DNA fragments if they have complementary nucleotide sequences. The combined construct can be directly delivered to competent cells, where the gaps and nicks between the fragments are completely rectified. We introduce a versatile sequence- and ligation-independent cloning (SLIC) method called 'DNA Assembly with Phosphorothioate (PT) and T5 Exonuclease' (DAPE), which generates precise lengths of 3' overhangs at both ends of linearized DNA. In contrast to conventional SLIC techniques, which are not suitable for cloning DNA fragments smaller than 50 base pairs (bp) due to overzealous exonuclease activity, such as with gRNA and epitope tags, DAPE can efficiently and precisely assemble several fragments in a single reaction regardless of the size of the DNA. Thus, DAPE, as an advanced toolkit for DNA cloning and synthetic biology, may further expedite the construction of more elaborate multi-gene circuitry.

## Introduction

Ligation-independent cloning (LIC) techniques have been widely used to easily and directly introduce multiple DNA fragments into a vector in a single experimental procedure with low cost. In comparison to conventional methods or other modular cloning techniques, such as Gateway cloning and Golden Gate cloning [1–3], SLIC allows any linearizable DNA fragment to be inserted into any position of any vector of choice without incorporating additional DNA sequences, a process known as 'seamless cloning'. In addition, SLIC allows multiplex assembly of DNA fragments in a single reaction. This capability dramatically reduces the time required for the cloning procedure. Thus, it can be regarded as a suitable method for applications in synthetic biology, such as the production of sophisticated gene circuitries based on Boolean logic gates.

**Funding:** This research was supported by the Basic Science Research Program through the National Research Foundation of Korea (NRF) funded by the Ministry of Education (NRF-2022R1A2C1012375).

**Competing interests:** The authors have declared that no competing interests exist.

Modern *in vitro* sequence-independent DNA assembly methods rely on generating relatively long complementary recessed ends (at least 12 bp) on DNA fragments. For the generation of single-stranded ends of DNA fragments, the linearized DNA can be reacted with enzymes that have exonuclease activity, such as T4 DNA polymerase [4], Klenow fragment [5], lambda exonuclease [6, 7], and T5 exonuclease [8–14], resulting in the production of stretched 5' or 3' overhangs, the size of which might be controlled by varying the reaction conditions. However, the precise length of overhangs cannot be guaranteed due to somewhat overzealous exonuclease activities of the enzymes. The uncontrollable nibbling activities of exonucleases could be a critical issue for cloning, especially when the DNA fragments are around 50 base pairs encoding gRNA for CRISPR/Cas9 or small epitope tags.

In this study, we introduce an alternative SLIC method, designated DAPE, which utilizes primers of PT internucleotide linkages to precisely generate the desired length of 3' single-stranded ends in PCR products. The PT internucleotide linkages have been reported to render DNA resistant to nuclease activities without affecting nucleotide base pairing [15, 16]. More interestingly, precisely measured 3' overhangs of 10 bp using PT linkages, by virtue of lambda exonuclease, facilitated large DNA integration into a specific DNA breakpoint generated by CRISPR/Cas9 [16]. We also found that incorporating five consecutive PT (5PT) linkages in the middle of primers makes the PTs near the ends of the PCR product highly resistant to T5 exonuclease activity. Because 3' overhangs cannot be produced by exonucleases with 3' to 5' gnawing activity, such as T4 DNA polymerase and the Klenow fragment, which leave 5' overhangs, we cannot apply them to DAPE. Since the T5 enzyme has been reported to have stronger exonuclease activity than lambda exonuclease, we ultimately chose the T5 enzyme for DAPE. This allows for the precise production of predetermined lengths of 3' recessed ends by T5 exonuclease. This approach offers a highly reliable and versatile method for the generation of recombinant DNA, regardless of the length of DNA fragments. Additionally, we further optimized the T5 exonuclease-based SLIC procedure, which performed superiorly to other exonuclease-based SLIC methods for both single insert and multi-piece cloning into a vector. DNA fragments as short as 50 bp, generated by PCR without undergoing a cleaning procedure, can be assembled in a single reaction with high precision and remarkable efficiency.

## Experimental procedures

### *E.coli* strain and plasmids

DH5α was cultured at 37°C in LB broth for the preparation of competent cells, which were titrated to reach $1x10^9$ cells per ml. The competent cells were prepared following the Inoue method [17]. 100 μl ($1x10^8$ cells) of the competent cells were used for transformation. The cloning vector pBluescript II SK+ (Agilent) and the expression vector pCS2+ were used for PCR amplification and linearized by digestion with XbaI, respectively. 10 ng of pBluescript II SK+ was used as the PCR template. pX330-puro was a gift from Sandra Martha Gomes Dias (Addgene plasmid # 110403; http://n2t.net/addgene:110403; RRID:Addgene 110403).

### Enzymes

T5 Exonuclease (M0663S), Lambda Exonuclease (M0262S), DNA Polymerase I, Large Klenow Fragment (M0210S), T4 DNA Polymerase (M0203S), T4 DNA Ligase (M0202S), and XbaI (R0145S) were purchased from New England Biolabs. T5 Exonuclease was 100-fold diluted before use in the T5 storage solution (S1 Table). Lamp Pfu DNA polymerase (BioFACT, LP116-250, Korea) was used for PCR. The DNA marker (TLD-204) was purchased from TransLab (Korea). All primers, including those with 5' phosphate and PT modifications, were purchased from BIONEER CORPORATION (Korea).

**Table 1. PCR condition for amplification of longer inserts.**

| PCR conditions | | | |
|---|---|---|---|
| Initial Denaturation | 95˚C | 2 min | 1 cycle |
| Denaturation | 95˚C | 20 sec | 25 cycles |
| Annealing | Tm | 40 sec | |
| Extension | 72˚C | 1 min/kb | |
| Final Extension | 72˚C | 5 min | 1 cycle |

## Gibson assembly

Commercially available Gibson assembly kit (AccuRapid™ Cloning Kit) wase purchased from BIONEER CORPORATION (Korea). DNA cloning was performed according to the manufacturer's instructions. Briefly, after the linear DNA mixture was reacted with the enzyme cocktail for 30 minutes at 50˚C, the reacted samples were transformed into DH5α competent cells.

## PCR condition

The sequences of primers used for PCR are listed in S2–S7 Tables. The PCR conditions for amplifying DNA over 110 bp are shown in Table 1. For the PCR, the following components were prepared at a final volume of 20 μl: 10 ng of template DNA together with 1 μl of each diluted primer (5 pmol per μl), 2 μl of 10x PCR reaction buffer, 0.4 μl of 10 mM dNTP mixture, and 0.2 μl of Lamp Pfu DNA polymerase (2.5 U per μl). Primer annealing PCR was performed to obtain DNA fragments of 80 bp or less, and the reaction conditions are shown in Table 2. For each reaction, 2 μl (5 pmol per μl) of primers, 2 μl of 10x PCR reaction buffer, 1 μl of 2.5 mM dNTP mixture, and 0.2 μl of Pfu DNA polymerase (2.5 U/μl) were used to make a final volume of 20 μl.

## Production of DNA overhangs

PCR samples were reacted with 1 U of diluted Lambda Exonuclease (NEB, 5 U per μl) and supplemented with 10x buffer provided by the manufacturer (NEB), making a total volume of 10 μl, and then incubated for 30 minutes at 37˚C. The composition of the Lambda Exonuclease dilution buffer is as follows: 25 mM Tris-HCl pH 8.0, 50 mM NaCl, 1 mM DTT, 0.1 mM EDTA, 50% glycerol. 1 U of diluted Klenow fragment (NEB, 5 U per μl) was used with 10x NEB buffer 2 to chew back the linear DNA from 3' → 5' direction. The mixture was incubated for 90 min at 37˚C before transformation. The composition of the Klenow fragment dilution buffer: 25 mM Tris-HCl pH 7.4, 1 mM DTT, 0.1 mM EDTA, 50% glycerol). The same amounts of DNA were reacted with 0.6 U of diluted NEB T4 DNA polymerase (3 U per μl), 10x NEB buffer 2.1 adjusted to 10 μl with distilled water. The samples were incubated for 1 minute at room temperature and then transferred onto ice for 10 minutes before transformation. The composition of the T4 DNA polymerase dilution buffer: 100 mM KPO₄ pH 6.5, 1 mM DTT, 50% glycerol. The reaction conditions of T5 exonuclease for DAPE are described in detail in

**Table 2. PCR condition for amplifying DNA shorter than 80 bp.**

| PCR conditions | | | |
|---|---|---|---|
| Initial Denaturation | 95˚C | 2 min | 1 cycle |
| Denaturation | 95˚C | 30 sec | 30 cycles |
| Annealing | Tm | 30 sec | |
| Extension | 72˚C | 30 sec | |
| Final Extension | 72˚C | 5 min | 1 cycle |

the Supplementary Information and S1 Table. Briefly, the same DNA samples were incubated with one hundredfold diluted T5 exonuclease (0.1 U) and 5x buffer for 30 min at 30°C on the heat block, followed by 8 min on ice before transformation.

## Transformation

The reaction samples were placed on ice for 8 minutes while the competent cells were thawed and then placed on ice. After 8 minutes of incubation on ice, 100 μl of competent cells was added to the sample and placed back on ice for another 8 minutes. The cells were heat shocked at 42°C in a water bath for 90 sec and then transferred to ice for 3 min. Afterward, 900 μl of LB broth was added to the samples and incubated at 37°C for 40 min. Subsequently, 100 μl of the transformants were spread on an LB plate containing the appropriate antibiotic and incubated until colonies became visible.

## Polyacrylamide gel electrophoresis

The 8% polyacrylamide gel was made with 1.86 ml of 30% acrylamide, 1.4 ml of 5x TBE buffer, 100 μl of 10% ammonium persulfate solution, 7 μl of TEMED, and 3.74 μl of Nuclease-Free water. 4 μl of PCR products (20%), with or without treatment of T5 exonuclease, were electrophoresed at 100 V for 1 hour, followed by staining with EtBr for 10 minutes. After destaining the gel in 1xTBE buffer four times for 10 minutes each at room temperature, the stained DNA was observed under the UV illumination.

## DNA sequencing

The randomly selected colonies were inoculated into 5 ml LB broth with appropriate antibiotics. The cells were grown overnight at 37°C, and the plasmid DNA was isolated using Exprep Plasmid SV (101–102, GeneAll, Korea) according to the manufacturer's instructions. Sequencing was performed by SolGent Analysis Service (Korea).

## Step by step DAPE protocol

1. Dilute T5 exonuclease one hundred times with T5 storage solution (10 μl of NEB T5 exonuclease (10 U) + 990 μl of T5 storage solution). The diluted enzyme must be stored at -20°C, and remains active for over six months. The composition of the storage solution is described in S1 Table.

2. Prepare 5x T5 reaction buffer. The composition of the 5x reaction buffer is described in S1 Table.

3. When using a vector as an acceptor, the vector should be linearized by RE digestion or amplified by PCR. The vector should be purified using a gene cleaning kit after gel electrophoresis. The detailed explanation of primer design for DAPE is shown in S4 Fig.

4. Prepare reaction mixture as follows:

| | |
|---|---|
| Linearized vector | 10 ~ 50 ng |
| Purified or crude PCR products | 1 μl (5% volume of PCR) or 50 ng |
| 5x T5 reaction buffer | 2 μl |
| 0.1 U T5 exonuclease | 1 μl |
| Distilled water | x μl |
| Total volume | 10 μl |

Note. When multi-piece DNA assembly is required, add the individual DNA fragments in the specified amounts as mentioned above, and then adjust the reaction volume to 20 µl with a two-fold increase of buffer and enzyme.

5. Place the reaction samples on the heat block for 30 minutes at 30˚C.

6. Note. Using a heat block is strongly recommended for DAPE.

7. After 30 min incubation on the heat block, place the samples on ice for 8 min.

8. Add 100 µl of competent cells to the 10 µl of reaction samples.

9. Perform transformation using the Inoue method.

## Results

### Direct comparison of SLIC techniques and optimization of T5 exonuclease-based SLIC

To test the efficiencies of currently available SLIC methods using enzymes with either 5' → 3' or 3' → 5' exonuclease activities. While T4 DNA polymerase and Klenow fragment exhibit only 3' → 5' exonuclease activities in the absence of dNTPs, resulting in 5' recessed ends, lambda exonuclease and T5 exonuclease can chew back in the 5' → 3' direction, leaving single-stranded 3' ends (S1 Fig).

We designed two primer sets to amplify approximately half of a pBluescript II SK+ vector each (Fig 1A). 50 ng of each of the purified PCR samples were used for testing cloning efficiency. The resulting two PCR products (~1.5 kb each) will have 15 bp complementary sequences at both ends, allowing them to anneal to each other. Consequently, the amplified products can form a circular structure only if their ends become single-stranded. The primers for PCR were designed in the following ways: 1. modified by adding a phosphate moiety at the 5' end (labeled P-15 mer-15 mer), 2. with a phosphate moiety at the 5' end and five consecutive PT linkages after the fifteenth nucleotide (P-15 mer-5PT-10 mer), 3. with five continuous PT linkages after the fifteenth nucleotide (15 mer-5PT-10 mer), 4. free of modifications (15 mer-15 mer) (Fig 1B and 1C). The lambda exonuclease showed the least cloning efficiency, regardless of primer modifications such as 5' phosphorylation. This modification was reported to be a prerequisite for the full activity of the exonuclease (Fig 1D and 1E) [18]. Though the other three enzymes showed impressive yields of colony numbers, T5 exonuclease was by far the best in terms of cloning efficiency, irrespective of primer modifications (Fig 1D and 1E). Therefore, we chose the T5 exonuclease for further optimization.

T5 exonuclease-based cloning was also known as TEDA (T5 exonuclease DNA assembly) [11]. To optimize the reaction conditions, we followed the same experimental procedures as in Fig 1, except using half a unit (0.1 U) of the T5 enzyme and reducing the reaction time from 40 minutes to 30 minutes. The reaction was carried out on a heat block at 30˚C. As shown in Fig 2, the cloning efficiency was higher under the conditions of using a lower unit of the enzyme with a reduced reaction time. As a negative control, we used an equal amount of linear DNA mixture without T5 exonuclease (-T5 Exo). Therefore, the data suggest that an excessive amount of T5 enzyme and a prolonged reaction period are detrimental to cloning. It is noteworthy that the reaction should be carried out on a heat block at 30˚C. When the samples were incubated in a conventional air-circulating incubator, the cloning efficiency was dramatically reduced, presumably due to ineffective heat transfer to the reaction tubes.

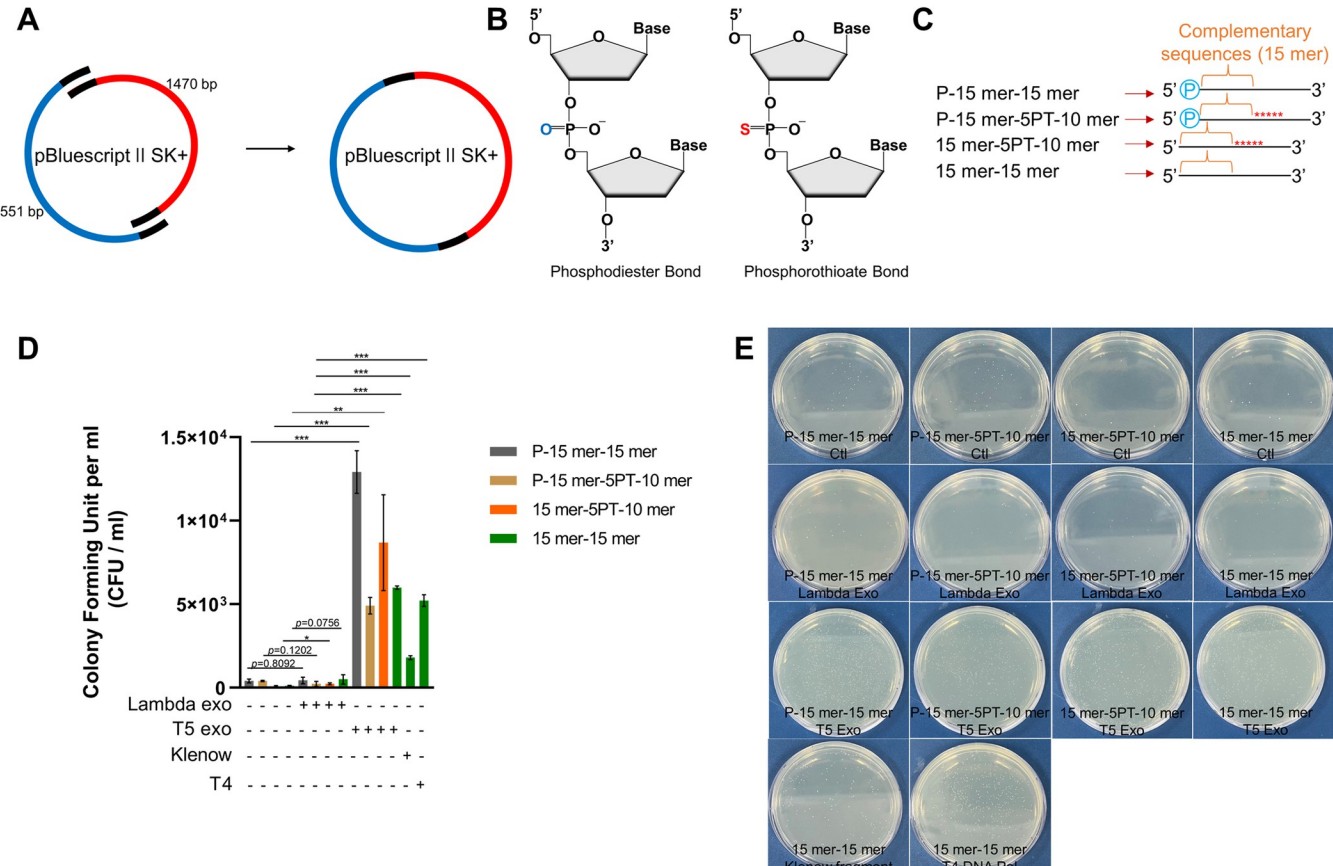

**Fig 1. Comparison of enzymes with exonuclease activities for SLIC experiments. (A)** A pBluescript II SK+ vector was chosen as a PCR template for amplifying the indicated lengths of fragments. Both PCR products, depicted in blue and red, were combined and assembled into the intact pBluescript II SK + vector using SLIC techniques. The black lines represent the 15 bp complementary sequences of each product. **(B)** A schematic diagram of phosphodiester and phosphorothioate (PT) bonds used in the preparation of primer synthesis. **(C)** Primers for amplifying the two different regions of pBluescript II SK+ in (A) were modified with a 5' end phosphate (P-15 mer-15 mer), a 5' end phosphate together with five PTs (P-15 mer-5PT-10 mer), five PTs only (15 mer-5PT-10 mer), or were not modified (15 mer-15 mer). The orange braces represent the complementary sequences. **(D)** Transformation efficiencies were calculated by counting the number of colonies in triplicate experiments. $1 \times 10^8$ competent cells were used for transforming the mixture of 50 ng of each purified PCR product after undergoing SLIC experiments. Single asterisk; P-values lower than 0.05, double asterisk; lower than 0.01, triple asterisks; P-value lower than 0.001. All experiments were repeated three times. **(E)** Representative plates of (D) are shown. CFU; colony forming unit, Ctl; control, Lambda Exo; lambda exonuclease, T5 exo; T5 exonuclease.

## Incorporating PT linkages in primers highly augmented the cloning efficiency of small PCR products

PT internucleotide linkages have been reported to protect linear DNA from the action of nucleases [15]. However, the incorporation of 5 PT linkages into the primers designed to produce 15 bp overhangs at both 3' ends of the PCR products did not elevate cloning efficiency (Fig 1D and 1E). We assumed that multiple PT linkages could only become a critical issue when the PCR products are as small as around 50 bp. To test our hypothesis, we designed several primer sets with identical sequences encoding a 6xHis epitope, varying only in the number of PT (from zero to five) linkages incorporated starting from the 16th nucleotide on the primer (Fig 3A). PCR was carried out after direct primer-to-primer annealing, and then 1 μl (5%) of the PCR products was used as inserts without undergoing any DNA cleaning procedure. The 48 bp PCR products were reacted with 0.1 U of T5 exonuclease for 30 minutes at 30°C on a heat block, and then separated on an 8% acrylamide gel. The PCR products with increased

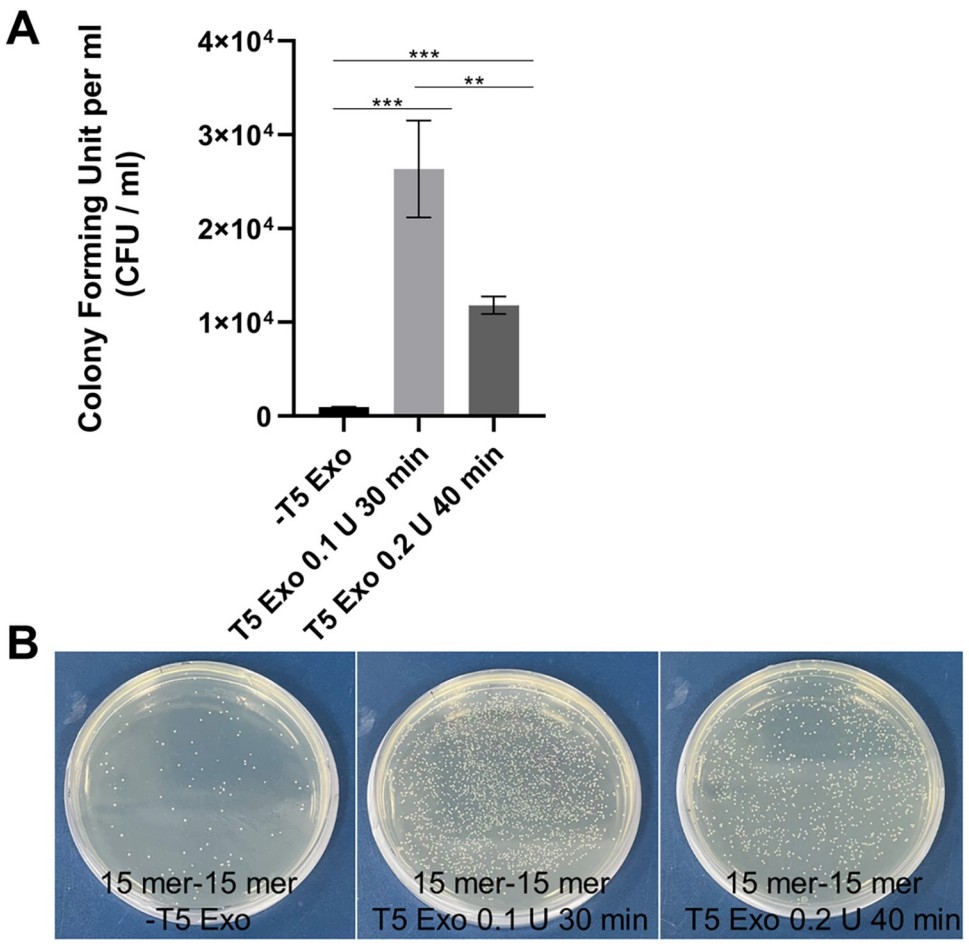

**Fig 2. Optimization of the T5-exonuclease-based SLIC method. (A)** Subcloning efficiency was increased by reducing the reaction time and the amount of T5 enzyme. The reaction was carried out at 30°C on a heat block. Double asterisks represent a P value less than 0.01, while triple asterisks indicate a P value less than 0.001, derived from three independent experiments. A linear DNA mixture without T5 exonuclease (-T5 Exo) was used as a negative control. **(B)** Representative plates of (B) are shown. CFU; colony forming unit.

numbers of PT linkages became gradually refractory to the T5 enzyme activities (Fig 3B). More importantly, the cloning efficiency of the PCR products into the linearized pCS2+ vector was dramatically augmented in the case of the inserts with five consecutive PT linkages in their primers (Fig 3C and 3D). The cloning accuracy of the method, called DAPE, was also validated by sequencing the clones (Fig 3E).

Next, we compared DAPE with Gibson assembly in terms of cloning efficiency and evaluated the practicality of both methodologies by incorporating 5' PT into the primer for cloning small DNA fragments. For this purpose, we used the same 48 bp PCR products from Fig 3 to directly compare DAPE and the commercially available Gibson assembly kit, employing primers with or without the incorporation of 5' PT. As expected, the linearized pCS2+ vector was unable to accommodate the 48 bp PCR fragments after treatment with T5 exonuclease or the Gibson assembly enzyme mixture (Fig 4). However, the cloning efficiency significantly increased when the PCR products were made refractory to T5 exonuclease activity through five consecutive PT modifications at their 5' ends, leaving 3' single-stranded DNA (Fig 4). More importantly, PT modifications in primers can be applied to Gibson assembly, as the

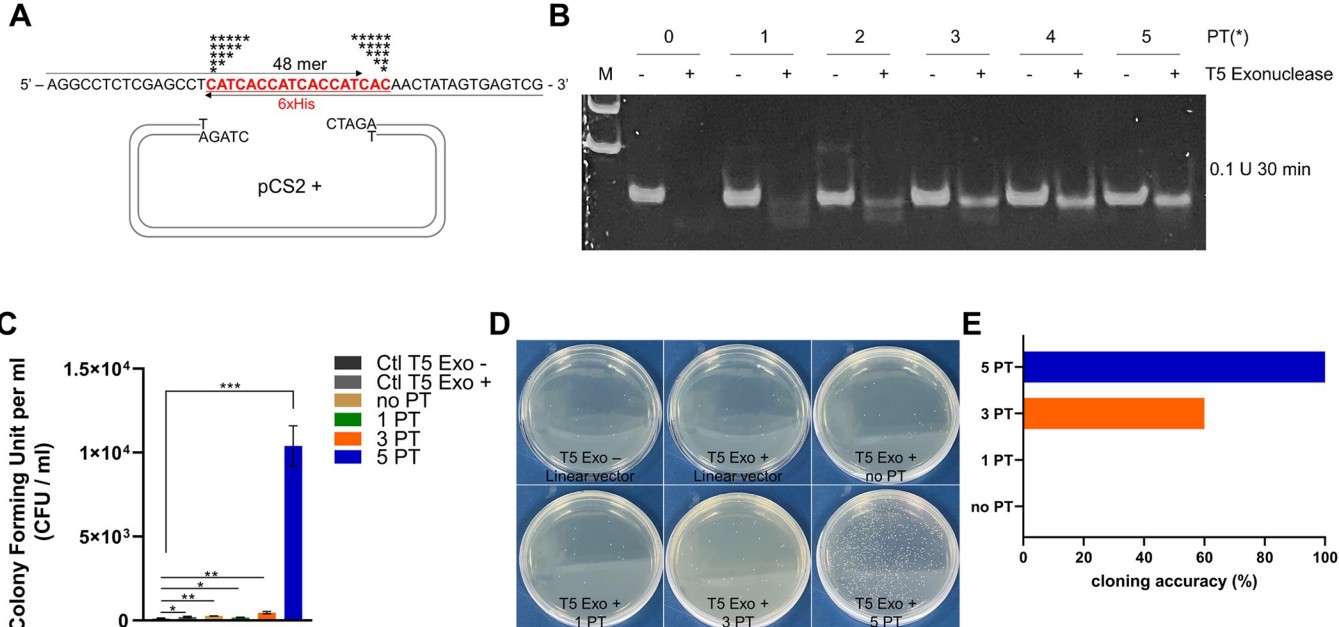

**Fig 3. At least 5 PTs are required for the subcloning of small DNA fragments. (A)** A schematic illustration of the subcloning procedure of a 6xHis epitope into an XbaI site of the pCS2+ vector. To amplify a 48 bp-length PCR product encompassing a 6xHis epitope (red uppercase), the indicated primers, depicted with arrows, were annealed via complementary sequences before undergoing PCR. The asterisks indicate the position and number of PT modifications. **(B)** Electrophoresis of PCR samples on an 8% acrylamide gel. The 4 μl of crude PCR products (20% of PCR) without purification were treated with or without 0.1 U of T5 exonuclease in a total reaction volume of 10 μl for 30 min at 30°C. Note that an increased number of PT internucleotide linkages proportionally stabilized the PCR products. M stands for the 100 bp size marker. **(C)** The PCR product with primers incorporating 5 PTs showed the highest subcloning efficiency into the pCS2+ vector. The experiments were carried out three times. Single asterisk; P-values lower than 0.05, double asterisk; lower than 0.01, triple asterisks; P-value lower than 0.001. **(D)** Representative images of (C). **(E)** Five randomly selected colonies from the plates in (D) were validated by Sanger sequencing. Only the samples with 5PT showed 100% subcloning accuracy. CFU; colony forming unit.

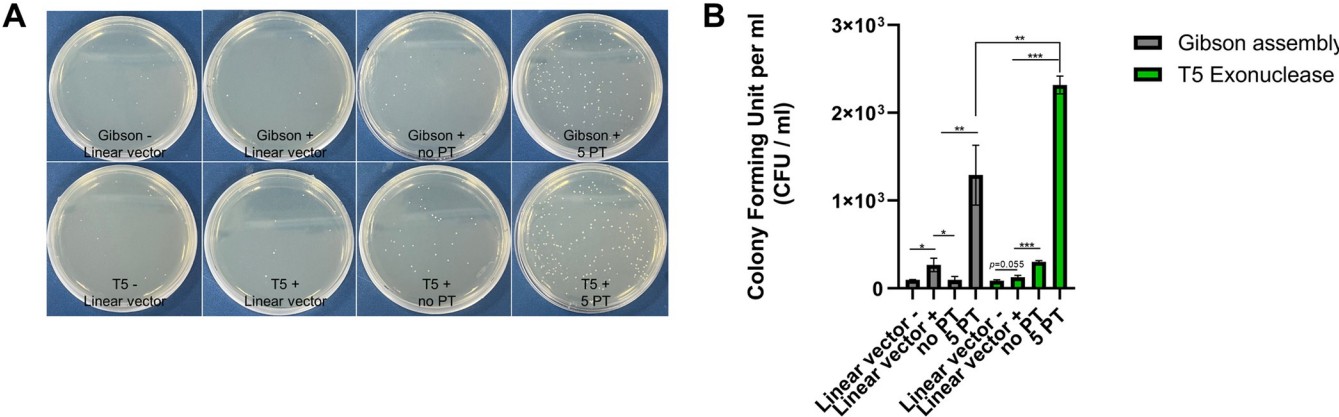

**Fig 4. Comparison of DAPE and Gibson assembly in terms of cloning efficiency for small DNA fragments. (A)** A 48 bp PCR product was subjected to DAPE or Gibson assembly, using primers with or without 5' PT modifications. A purified 50 pg of pCS2+ vector, linearized by Xba I digestion, was used for cloning with 1 μl of crude PCR samples. Gibson-/linear vector; enzyme free reaction with linearized vector only. Gibson+/linear vector; Gibson technology reaction with linearized vector only. Gibson+/no PT; Gibson technology reaction with linearized vector and PCR amplified insert without 5PT modification. Gibson+/5PT; Gibson technology reaction with linearized vector and 5PT modified PCR product. T5-/linear vector; T5 free reaction with linearized vector only. T5+/linear vector; T5 (0.1 U) reaction with linearized vector only. T5+/no PT; T5 (0.1 U) reaction with linearized vector and PCR amplified insert without 5PT modification. T5+/5PT; T5 (0.1 U) reaction with linearized vector and 5PT modified PCR product. **(B)** Graphical representation of the data shown in (A). Single asterisk; P-values lower than 0.05, double asterisk; lower than 0.01, triple asterisks; P-value lower than 0.001. All experiments were repeated three times. CFU; colony forming unit.

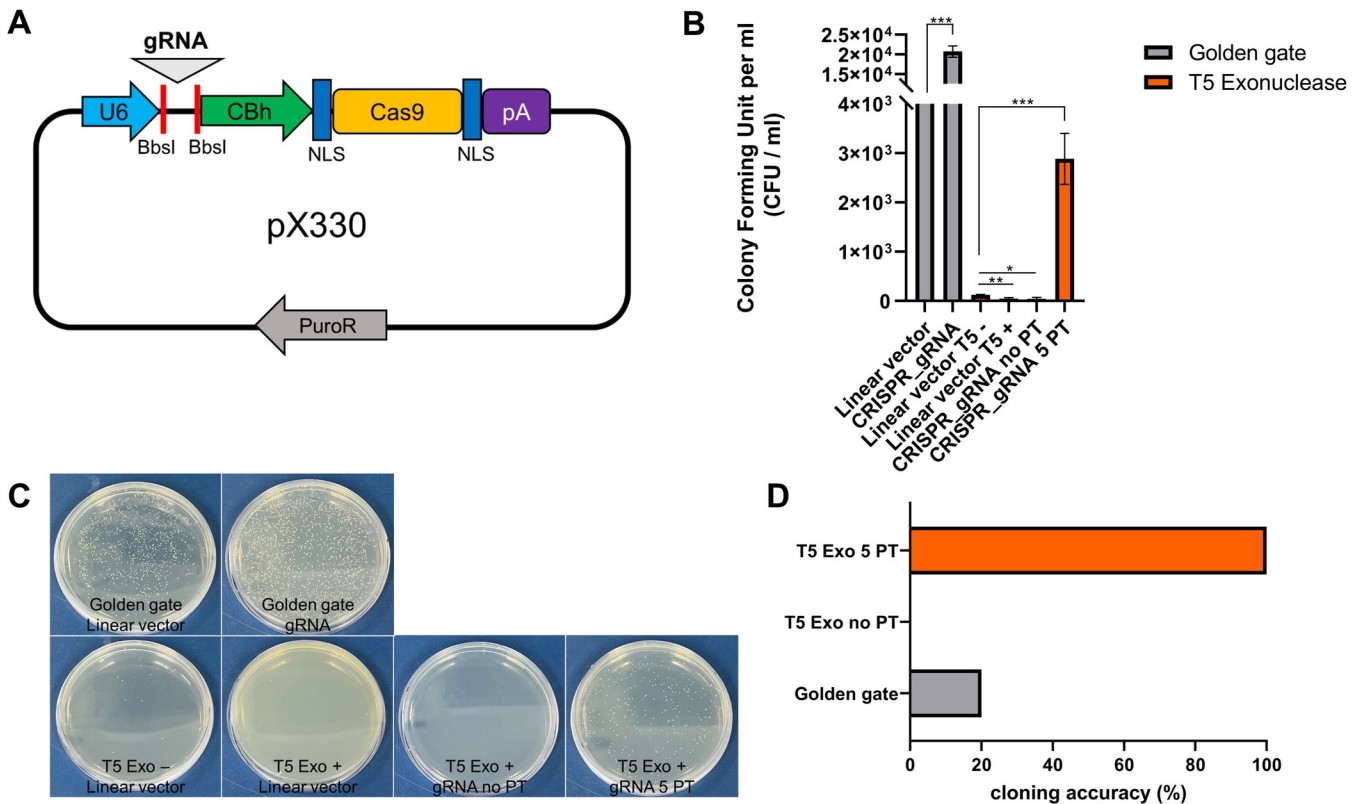

**Fig 5. gRNA cloning for CRIPR/Cas9. (A)** A schematic diagram of gRNA cloning into the px330-puro vector. After linearization with BbsI treatment, 50 ng of the linear vector was subjected to cloning procedures with 1 μl (5%) of 54 bp PCR products for DAPE or the same amount of 52 bp PCR products for Golden Gate cloning. While the linearized vector was purified using purification kits, the crude PCR products were directly used for cloning. **(B)** Comparison of Golden Gate cloning and DAPE. Although Golden Gate cloning (left) yielded more colonies than DAPE (right), the former also resulted in a higher number of non-positive clones. DAPE without PTs failed to form positive clones. A linear vector without gRNA was used as a negative control (labeled as "linear vector"). Single asterisk; P-values lower than 0.05, double asterisk; lower than 0.01. The triple asterisks represent P value lower than 0.001, derived from triple experiments. **(C)** Representative individual images of triplet (B). **(D)** Cloning accuracies of Golden Gate cloning and DAPE were validated by sequencing. Five randomly selected clones were sequenced to identify the orientation and sequences of the gRNA. Only DAPE with 5 PTs showed 100% accuracy of cloning. CFU; colony forming unit.

cloning efficiency of small DNA fragments using Gibson assembly was also significantly increased, though slightly less effective compared to DAPE (Fig 4).

Due to the high yield of positive colonies when exploiting T5 exonuclease together with PT linkages, we wondered whether the DAPE method could be applied to guide RNA (gRNA) cloning for CRISPR/Cas9. We designed a gRNA targeting BRAF exon 1 to insert between two BbsI recognition sites of a px330.puro vector (Fig 5A, S2 Fig). In comparison with Golden Gate cloning, DAPE showed high accuracy with significantly fewer self-ligated clones (Fig 5B–5D). As expected, without PT engagement, we could not succeed in gRNA cloning using T5 enzyme (Fig 5B and 5C). There are significant background clones in Golden Gate cloning (Fig 5B and 5C), which could be due to the use of T4 DNA ligase. Therefore, avoiding the use of DNA ligase during DNA assembly would be preferable for increasing the proportion of positive clones.

## Evaluation of insert size for DAPE

Next, to evaluate the optimal size of DNA fragments to be amplified with a PT-modified primer set for DAPE cloning, we tested different lengths of PCR products, starting from 50 bp

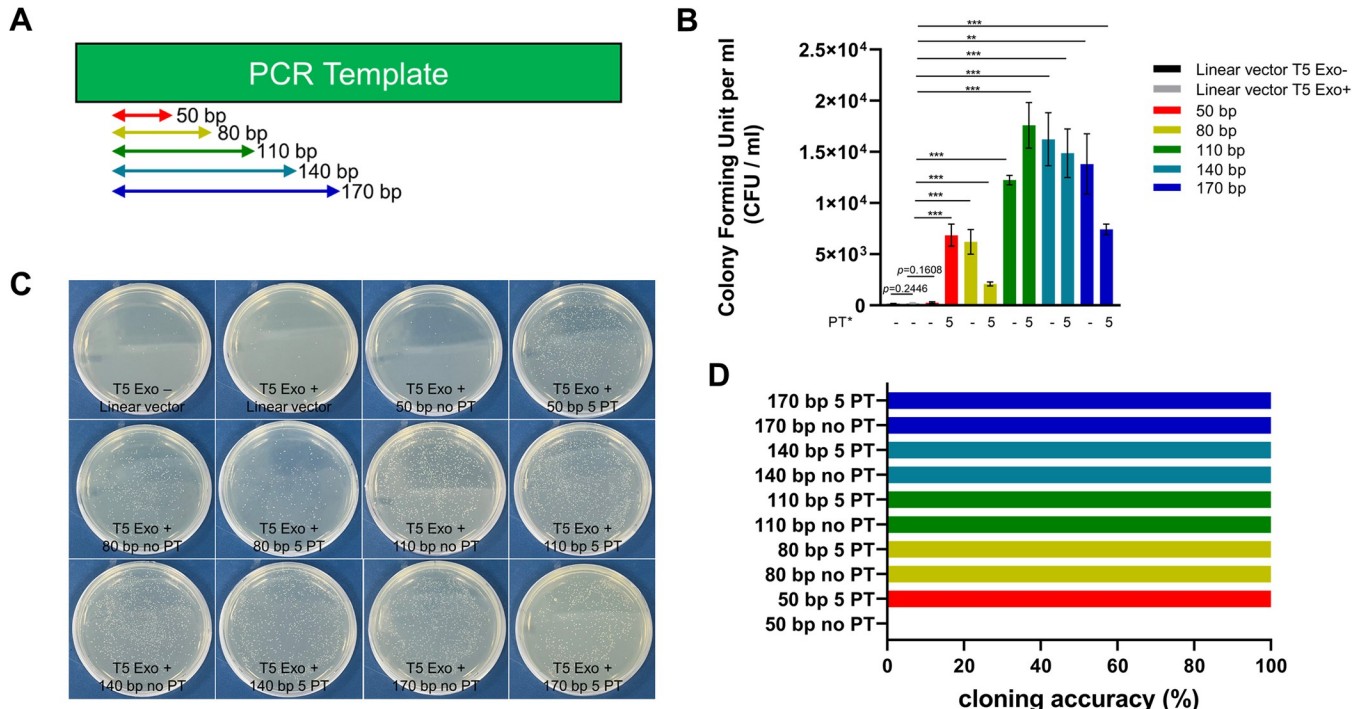

**Fig 6. Five PTs are required when the DNA fragments are smaller than 80 bp. (A)** A schematic diagram of the experimental procedure. Different sizes of DNA fragments encoding a portion of PCR template (EGFP) were amplified with the indicated primers depicted by arrows. The red and blue lines attached to the primers represent the 15 bp complementary sequences between the pCS2+ vector and the insert. The pCS2+ vector was linearized with XbaI before purification. 50 ng of linearized vector was used for subcloning with the same amount of DNA, except for a 50 bp and 80 bp inserts, of which 1 μl (5%) of crude PCR products was directly used for DAPE cloning. **(B)** Five PTs are obligatory for cloning when the DNA fragments are smaller than 50 bp, and are highly recommended when they are less than 80 bp. Double asterisks represent a P value less than 0.01, while triple asterisks indicate a P value less than 0.001, derived from three independent experiments. **(C)** Representative plates of triplet (B). **(D)** Five randomly selected clones from the plates in (C) were sequenced to evaluate the accuracy of DAPE. Except for clones obtained from samples without 5 PTs of 50 bp inserts, all others showed 100% accuracy in cloning. CFU; colony forming unit.

and increasing by 30 bp increments up to 170 bp of the EGFP portion (Fig 6A). For precision, we fixed the forward primer for PCR amplification (Fig 6A). The individual PCR products were subcloned into the pCS2+ vector, which was linearized by digestion with XbaI. While 1 μl (5%) of small PCR product (50 bp and 80 bp) was directly added to the reaction tube, 50 ng of other gene-cleaned PCR products were used for cloning with 50 ng of the linear vector. As shown in Fig 6B and 6C, the PCR products over 80 bp did not require PT linkage to increase cloning efficiency. However, the PCR product of 50 bp without five PTs could not be subcloned into the linearized vector (Fig 6B and 6C). Except for clones that emerged from transformants with non-PT PCR products of 50 bp, all other randomly selected clones were accurately oriented and free of mutations. These clones were validated by sequencing (Fig 6D). Therefore, we suggest that when the PCR products are smaller than 80 bp, five PTs should be incorporated into the primers for DAPE cloning.

## 5PT may inhibit the formation of secondary structures in the 3' overhang

To evaluate the potential advantages of 5PT insertion into primers for sequences likely to form secondary structures, we utilized a 30 bp segment of the leadzyme sequence, which is well-known for its propensity to form secondary structures (Fig 7A and 7B) [19, 20]. Compared to the subcloning efficiency of TEDA and Gibson assembly using unmodified primers, PCR-amplified DNA fragments (~1 kb) with primers containing 5PT were efficiently incorporated

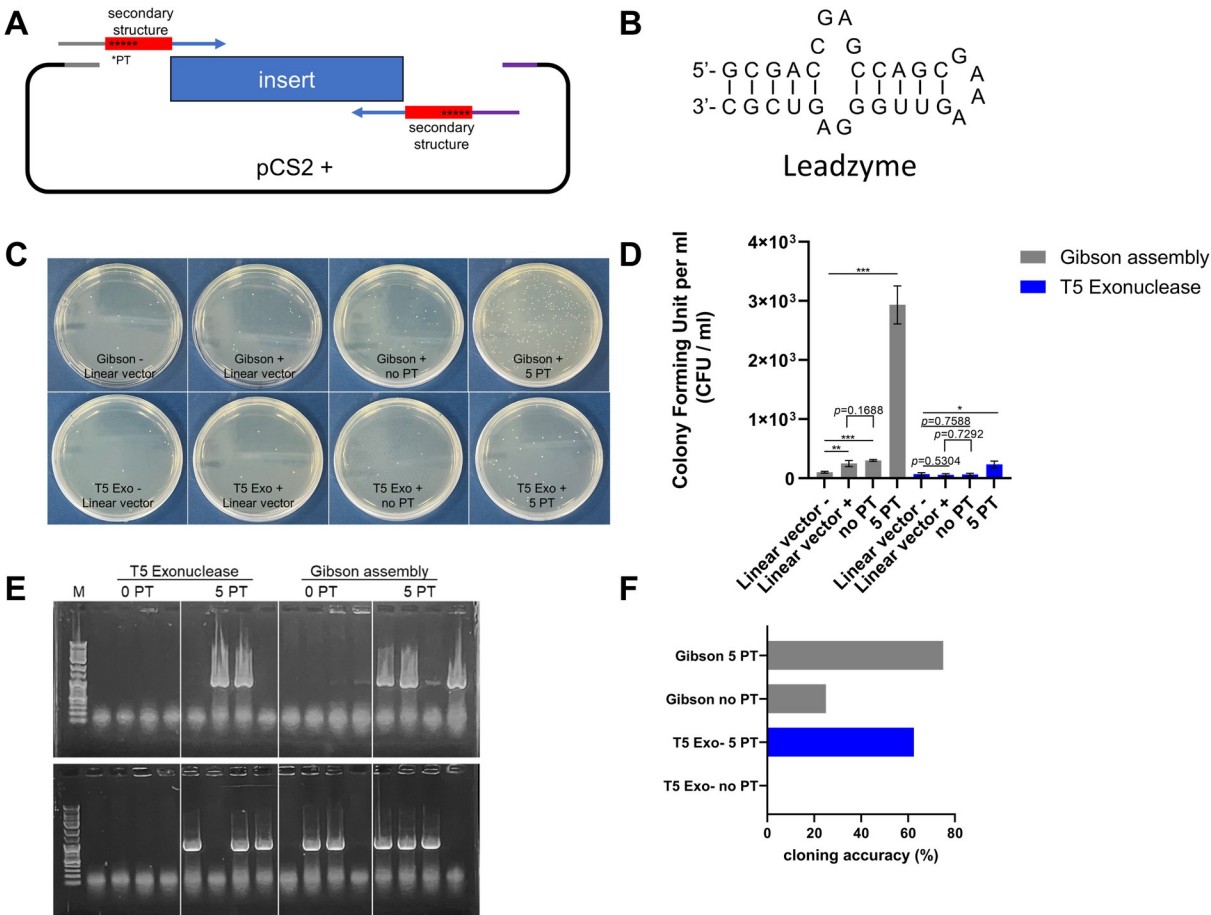

**Fig 7. 5PT-incorporated primers may prevent secondary structure formation at both ends of PCR products after reaction with T5 exonuclease.** (**A**) A schematic diagram of the experimental condition. The linearized pCS2+ vector, digested with XbaI, was reacted with an insert encoding a kanamycin resistance gene and containing 30-bp leadzyme sequences at both ends, which are prone to forming secondary structures. (**B**) Leadzyme sequence and its expected secondary structure. (**C**) Representative plates showing subcloning results. Gibson+/linear vector; Gibson technology reaction with linearized vector only. Gibson+/no PT; Gibson technology reaction with linearized vector and PCR amplified insert without 5PT modification. Gibson+/5PT; Gibson technology reaction with linearized vector and 5PT modified PCR product. T5-/linear vector; T5 free reaction with linearized vector only. T5+/linear vector; T5 (0.1 U) reaction with linearized vector only. T5+/no PT; T5 (0.1 U) reaction with linearized vector and PCR amplified insert without 5PT modification. T5+/5PT; T5 (0.1 U) reaction with linearized vector and 5PT modified PCR product. (**D**) Graphical and statistical representation of the data shown in (C). Single asterisk; P-values lower than 0.05, double asterisk; lower than 0.01, triple asterisks; P-value lower than 0.001. All experiments were repeated three times. (**E**) Colony PCR analysis of the emerging clones on the plate, performed on a 1% agarose gel. Eight randomly selected colonies were subjected to PCR using primers that amplify the expected DNA size (~1 kb). (F) Graphical representation of the date shown in (E). CFU; colony forming unit.

into the linearized vector in both DAPE and Gibson assembly (Fig 7C–7F). Among eight randomly selected colonies from each plate (Fig 4C), the PCR products with 5PT in both DAPE and Gibson assembly were inserted into the vector with over 62% efficiency, compared to conventional Gibson assembly (2 out of 8) and TEDA (none out of 8) (Fig 7E and 7F). Therefore, the data suggest that incorporating 5PT from the 16th base of the 5'-primer may prevent further nibbling activity of T5 exonuclease and inhibit secondary structure formation at both single-stranded ends of the PCR product. It is noteworthy that, although Gibson assembly with 5PT appeared superior to DAPE, the Gibson cloning method also increased the occurrence of self-ligated products (labeled as 'Gibson + linear vector'), likely due to the use of Taq ligase, as previously reported (Fig 7D) [8, 10, 11].

## Assembly of multiple DNA fragments with DAPE

To investigate whether the DAPE method could be applied to connect multiple small DNA fragments in a designated order, we attempted to attach two small epitopes, 6xHis and Flag, to the ends of EGFP to be assembled within a linearized pCS2+ vector (Fig 8A). While the 6xHis (54 bp) and Flag (57 bp) sequences were PCR-amplified with or without five-PT incorporated primers (S3 Fig), the EGFP was prepared free of PTs. The crude, tiny-sized PCR products were combined with purified EGFP and XbaI-digested pCS2+ vector. This mixture was reacted with T5 exonuclease (0.1 U) in a single tube for 30 minutes at 30°C and then transformed into competent cells. Although the overall transformation efficiency was dramatically reduced by increasing the number of inserts, the PT-protected PCR products were still better aligned with EGFP in comparison to the reaction with the non-PT PCR samples in the pCS2+ vector (Fig 8B and 8C). We could confirm the PCR reactions by observing the size difference above the primer dimers (Fig 8D). The higher accuracy of multiple DNA assembly with DAPE compared to SLIC without PT was also validated by sequencing. While only one clone among ten was correctly assembled in conventional SLIC, 40% of clones produced by DAPE were accurately aligned in the correct orientation (Fig 8E).

## Discussion

SLIC technology has several advantages over conventional restriction enzyme (RE) and ligase-dependent cloning methodologies: first, SLIC is free from concerns about RE sites; second, it

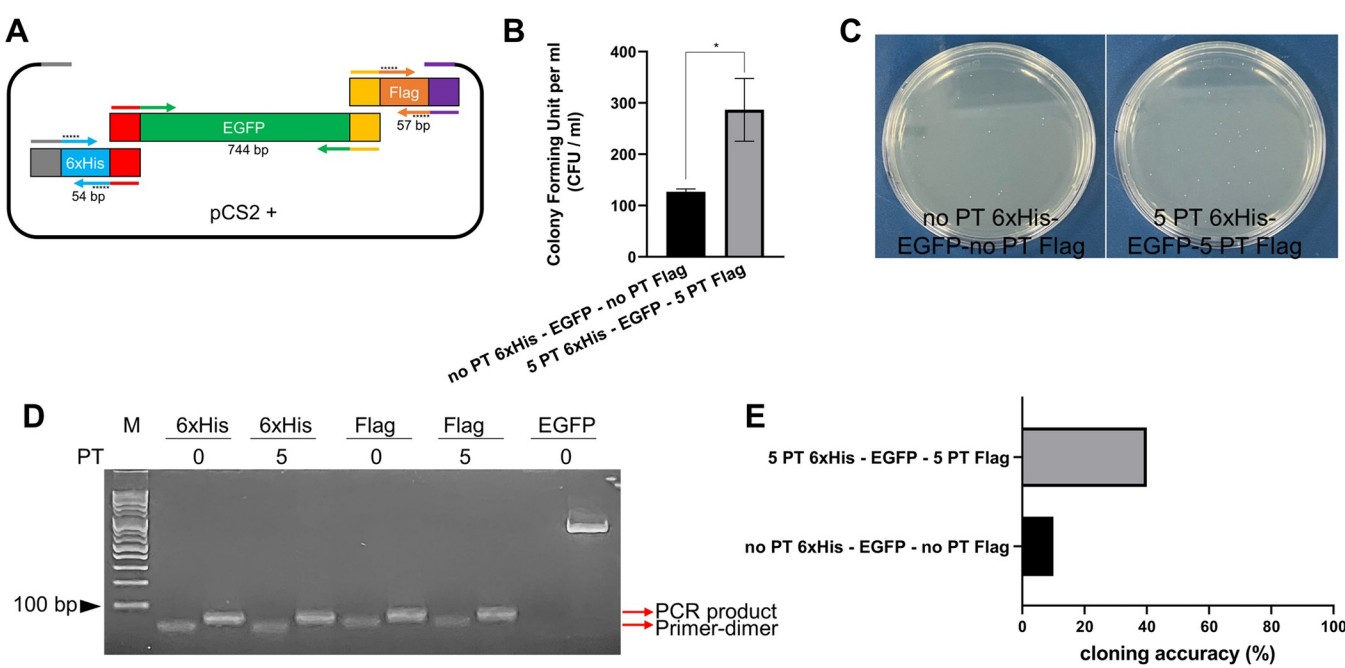

**Fig 8. Multiple DNA fragments assembly with DAPE. (A)** A schematic illustration of the DNA assembly. The identically colored ends of DNA fragments represent 15 bp complementary sequences. Asterisks on the primers represent 5PT linkages. The pCS2+ vector was linearized with XbaI, and then 50 ng of the vector was used for the DNA assembly experiment. The 54 bp length of 6xHis and the 57 bp length of Flag epitope were amplified with 5 PTs primers. The full-length EGFP was amplified with non-PT primers. While 50 ng of EGFP PCR product was used for DNA assembly, 1 μg (5%) of each crude small DNA epitope was added to the reaction tube for DAPE. **(B)** Comparison of DAPE with conventional SLIC using T5 exonuclease. The single asterisk represents P-values lower than 0.05, derived from triple DNA assembly experiments. **(C)** Representative images of (B) from among the triple experiments. **(D)** Electrophoresis of PCR products on a 2% agarose gel. M represents 100 bp size marker. The small epitopes amplified by PCR migrated slower compared to the primer dimers. **(E)** Ten randomly selected clones from the plates in (C) were validated by sequencing to analyze the accuracy of DNA assembly. DAPE is superior in both accuracy and efficiency compared to conventional SLIC when using small DNA fragments in multiple DNA assembly. CFU; colony forming unit.

does not require DNA ligase; third, it maintains the directionality of the insert; fourth, it helps avoid the integration of unwanted sequences; fifth, it enables the assembly of multiple DNA fragments in a single reaction. However, one major concern of SLIC, including Gibson assembly, is its reliance on the intrinsic exonuclease properties of DNA polymerases or exonucleases. These methods suffer from uncontrollable enzyme activity, making it impossible to produce a predetermined length of single-stranded DNA.

We chose T5 exonuclease to develop a novel SLIC method because, compared with several other enzymes used for SLIC, T5 demonstrated superior cloning efficiency. Through optimization, we adjusted the optimal number of PTs on primers for both single-piece DNA cloning and multi-piece DNA assembly. Five PTs prevented the PCR products from being degraded by the exonuclease activity of the T5 enzyme, resulting in the production of the designated length of 3' overhangs.

There are several alternative SLIC methods for producing precisely defined length of single-stranded overhangs. The initial attempt of LIC, known as LIC-PCR, aimed to produce 12 bp complementary overhangs using both the DNA polymerase activity and exonuclease property of T4 DNA polymerase [21]. However, a flaw of LIC-PCR is that the method imposes nucleotide restrictions on the sequences that are supposed to be single-stranded overhangs. Another technique to precisely produce 12 bp overhangs involves using modified primers, also known as terminator primers, with ribonucleotides incorporated at positions 13 to 15 [22]. By exploiting a specific trait of Pfu, which does not use RNA as a template, the PCR products amplified using terminator primers acquire single-stranded overhangs. The disadvantage of terminator primer-mediated LIC is the high cost associated with inserting ribonucleotides into primers and the relatively high propensity for mutations at the cloning junction. Alternatively, ribonucleotides can be cleaved to produce 3' overhangs by incubating the PCR products with lanthanum cation ($La3^+$) or lutetium ion ($Lu3^+$) [23]. PT-based generation of 12 bp 3' overhangs was also reported previously [24, 25]. The PT linkage becomes unstable under alkaline conditions supplemented with iodine and ethanol, causing PCR products with multiple PTs at their 5' ends to lose PT-linked nucleotides under such chemical conditions [24, 25]. However, since the length of the overhangs strictly depends on the number of PTs, the cost of primers would increase proportionally with the length of the 3' overhangs. Another alternative to generate predetermined sizes of overhangs is the twin-primer assembly method (TPA), which uses only DNA polymerase to amplify DNA with long sticky overhangs [26, 27]. TPA would be advantageous when cloning small DNA fragments because the amplified DNA is not subjected to enzymes with exonuclease activity. However, to produce overhangs with TPA, two separate PCR products must be denatured and re-annealed, resulting in only a quarter of the mixture producing the desired length of overhangs. TPA appears to be a possible option for synthesizing assembled DNA with up to 10 fragments. We suggest that DAPE could be an alternative method, simpler than TPA, for producing a designated length of 3' overhangs for DNA assembly.

Small DNA fragments, such as gRNA, can be subcloned directly into the vector using a simple primer annealing method after designing the primers with appropriate overhangs. However, the method requires DNA ligase for cloning, which inevitably results in background colonies similar to those produced by Golden Gate cloning (Fig 5). This makes it cumbersome to isolate the desired clones. In addition, if the vector is linearized with a single RE at a site, the insert generated by primer annealing can be integrated randomly, without proper directionality. Thus, since DAPE, like the conventional SLIC methods, does not rely on DNA ligase while maintaining directionality, it could be an alternative technique capable of replacing the primer annealing-mediated subcloning methodology.

DAPE is a modified technique derived from TEDA, which is a simplified method of Gibson assembly and Hot Fusion [8, 10, 11]. Based on our results, we suggest that when the insert DNA is shorter than 80 bp, the primers should be designed to have five consecutive PTs immediately following the sequences to be erased by T5 enzyme. We do not recommend adding PTs to primers when the DNA fragments are longer than 80 bp. We provided the details of the primer design for DAPE in S4 Fig. By manipulating nucleotide sequences, we can add or delete RE sites between cloning junctions (S4 Fig). We also provided a detailed protocol for DAPE, along with the compositions of an T5 storage solution and an optimized 5x reaction buffer (Supplementary Information). The T5 exonuclease (0.1 U/μl), stored in the T5 storage solution at a hundredfold dilution, remained active for over six months.

The DAPE cloning methodology could be expanded to the LOCK technology, which was developed for the targeted insertion of a large DNA fragment into the mammalian genome [16]. The most critical point of LOCK is to generate a 10 bp 3' overhang, where the complementary sequence of the unfolded genomic region binds through the helicase activity of Cas9. According to LOCK technology, the length of the 3' overhang should be 10 bp for the most efficient knock-in of relatively large DNA fragments [16]. Thus, we propose that DAPE could be an alternative to LOCK for producing a 10 bp 3' overhang, in which lambda exonuclease is used instead of the T5 enzyme.

In comparison to Gibson assembly, the potential advantages and disadvantages of DAPE are as follows: First, by omitting DNA ligases, researchers can suppress the emergence of self-ligated clones, albeit at the cost of reducing the overall number of colonies, as shown in Figs 5 and 7. Previous reports indicate that while Gibson-based cloning, which includes Taq ligase, caused relatively high levels of background clones, other simplified Gibson techniques, such as Hot Fusion and TEDA, which are free of DNA ligase, produced fewer self-ligated clones without affecting cloning efficiency [8, 10, 11]. Second, we can achieve optimal results by using 5PT-incorporated primers in combination with Gibson assembly to subclone a DNA that contains sequences prone to forming secondary structures at both ends (Fig 7). There are two rational explanations for the improved cloning efficiency of 5PT with Gibson assembly. First, 5PT may protect sequences located more than 15 bp from the 5' end, preventing the remaining 3' single-strand from forming a hairpin structure. Second, while DAPE was performed at 30˚C, Gibson assembly was conducted at 50˚C, a relatively high temperature that likely helps prevent secondary structure formation. Considering the results in Fig 4, DAPE is a preferable choice for constructing plasmids with small DNA fragments. However, under more complex conditions, such as concerns about secondary structure formation at the ends of the DNA fragment, Gibson assembly with 5PT-modified primers can enhance overall cloning efficiency. Additionally, it is important to note that the 5PT modification was not advantageous for relatively long DNA fragments, as shown in Fig 1. Therefore, we suggest that 5PT should be applied to DNA fragments shorter than 80 bp, as demonstrated in Fig 6.

Since the cloning efficiency of multi-piece DNA assembly by TEDA-based SLIC gradually increased with longer complementary single-stranded overhangs [11, 14], the relatively low yield of DAPE for assembling multiple DNA fragments could be improved by extending the 3' overhang to over 20 bp. Alternatively, further optimization of DAPE could be helpful in increasing the success rate of multi-piece DNA assembly. A recent report suggested that TLTC is an efficient alternative for leaving 15 bp of 3' overhang. This method involves incubation at low temperature (0˚C) with a high concentration of T5 exonuclease (0.5 U) for 5 minutes [14]. It would be feasible to combine DAPE with TLTC to evaluate whether it improves cloning efficiency. Thus, there is still ample room for improving DAPE to make it more suitable for DNA assemblies, a key recombination technique in synthetic biology. Collectively, we suggest that DAPE and Gibson assembly with 5PT could serve as valuable alternative techniques for

cloning intractable DNA, particularly DNA forming secondary structures and small DNA fragments encoding gRNA, small domains, or epitopes.

## Supporting information

**S1 Fig. Schematic diagrams of the exonuclease activities of various enzymes.** (A) T4 DNA polymerase and the large Klenow fragment, without supplementation of dNTPs, exhibit only 3' →5' exonuclease function, producing single-stranded DNA at the 5' end. (B) T5 and lambda exonucleases have 5' →3' exonuclease activities, resulting in the production of 3' overhangs. (C) Linear DNA with five tandem PTs defies the activities of exonuclease. Asterisks indicate PTs.
(TIF)

**S2 Fig. To amplify a 54 bp or 52 bp DNA fragment encompassing BRAF targeting gRNA, the primers depicted with arrows can form dimers via their complementary sequences.** The primer dimers underwent PCR. Asterisks indicate the position and number of PT modifications, while red capitals represent gRNA sequences.
(TIF)

**S3 Fig. PCR using only annealed primers was conducted to amplify the 54 bp 6xHis tag and the 57 bp Flag epitope.** Asterisks represent PT-modified nucleotides, while red uppercase letters indicate sequences encoding 6xHis or Flag epitope.
(TIF)

**S4 Fig. Primer design for DAPE.** (A) The linearized vector, digested with the indicated REs (EcoRI, BamHI producing 5' overhangs; ApaI producing 3' overhangs; and EcoRV producing blunt ends), serves as the acceptor of the PCR products. Arrows represent primers, while dotted boxes denote the 15 bp complementary sequences between the termini of the vector and the PCR product. N stands for the sequences specific to the desired insert. Five consecutive asterisks above the blue 'N' letters represent the positions of PT internucleotide linkages. After subcloning into the vector, the RE recognition sites may inevitably be erased. (B) The RE sites can be restored by adding appropriate nucleotides to the primers. The red capital letters represent the sequences for the indicated RE sites.
(TIF)

**S1 Table. Composition of the T5 storage solution and the 5x T5 reaction buffer.**
(PDF)

**S2 Table. List of primers used for the experiments in Figs 1 and 2.**
(PDF)

**S3 Table. List of primers used for the experiments in Figs 3 and 4.**
(PDF)

**S4 Table. List of primers used for the experiments in Fig 5.**
(PDF)

**S5 Table. List of primers used for the experiments in Fig 6.**
(PDF)

**S6 Table. List of primers used for the experiments in Fig 7.**
(PDF)

**S7 Table. List of primers used for the experiments in Fig 8.**
(PDF)

**S1 Raw images. The raw agarose gel electrophoresis images as seen under a UV transillu-minator.**
(PDF)

## Acknowledgments

The authors gratefully acknowledge Prof. Hangil Lee (Chungnam Nat. Univ.) and Prof. Seok-Yong Choi (Chonnam Nat. Univ.) for thoroughly reading the manuscript and providing valu-able comments.

## Author Contributions

**Conceptualization:** Seoee Lee, Hyunju Ro.

**Data curation:** Seoee Lee.

**Formal analysis:** Seoee Lee.

**Funding acquisition:** Minho Won, Hyunju Ro.

**Investigation:** Seoee Lee.

**Methodology:** Seoee Lee.

**Project administration:** Seoee Lee.

**Supervision:** Minho Won, Hyunju Ro.

**Validation:** Hyunyoung Kim,  Aqsa, Kwangjin Jeung.

**Writing – original draft:** Seoee Lee, Hyunju Ro.

**Writing – review & editing:** Seoee Lee, Hyunju Ro.

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
