## [Decision Letter · Decision Letter 0]

28 Oct 2024

PONE-D-24-43146DAPE Cloning with Modified Primers for Producing Designated Lengths of 3’ Single-Stranded Ends in PCR ProductsPLOS ONE

Dear Dr. Ro,

Thank you for submitting your manuscript to PLOS ONE. After careful consideration, we feel that it has merit but does not fully meet PLOS ONE’s publication criteria as it currently stands. Therefore, we invite you to submit a revised version of the manuscript that addresses the points raised during the review process.

We look forward to receiving your revised manuscript.

Kind regards,

Zheng Yuan

Academic Editor

PLOS ONE

Journal Requirements:

4. Thank you for stating the following financial disclosure: “This research was supported by the Basic Science Research Program through the National Research Foundation of Korea (NRF) funded by the Ministry of Education (NRF-2022R1A2C1012375).”

5. We note that your Data Availability Statement is currently as follows: “All relevant data are within the manuscript and in Supporting Information files.”

Please confirm at this time whether or not your submission contains all raw data required to replicate the results of your study. Authors must share the “minimal data set” for their submission. PLOS defines the minimal data set to consist of the data required to replicate all study findings reported in the article, as well as related metadata and methods (https://journals.plos.org/plosone/s/data-availability#loc-minimal-data-set-definition). For example, authors should submit the following data: - The values behind the means, standard deviations and other measures reported; - The values used to build graphs; - The points extracted from images for analysis. Authors do not need to submit their entire data set if only a portion of the data was used in the reported study. If your submission does not contain these data, please either upload them as Supporting Information files or deposit them to a stable, public repository and provide us with the relevant URLs, DOIs, or accession numbers. For a list of recommended repositories, please see https://journals.plos.org/plosone/s/recommended-repositories. If there are ethical or legal restrictions on sharing a de-identified data set, please explain them in detail (e.g., data contain potentially sensitive information, data are owned by a third-party organization, etc.) and who has imposed them (e.g., an ethics committee). Please also provide contact information for a data access committee, ethics committee, or other institutional body to which data requests may be sent. If data are owned by a third party, please indicate how others may request data access.

7. PLOS ONE now requires that authors provide the original uncropped and unadjusted images underlying all blot or gel results reported in a submission’s figures or Supporting Information files. This policy and the journal’s other requirements for blot/gel reporting and figure preparation are described in detail at https://journals.plos.org/plosone/s/figures#loc-blot-and-gel-reporting-requirements and https://journals.plos.org/plosone/s/figures#loc-preparing-figures-from-image-files. When you submit your revised manuscript, please ensure that your figures adhere fully to these guidelines and provide the original underlying images for all blot or gel data reported in your submission. See the following link for instructions on providing the original image data: https://journals.plos.org/plosone/s/figures#loc-original-images-for-blots-and-gels. In your cover letter, please note whether your blot/gel image data are in Supporting Information or posted at a public data repository, provide the repository URL if relevant, and provide specific details as to which raw blot/gel images, if any, are not available. Email us at plosone@plos.org if you have any questions.

Additional Editor Comments:

Reviewer 1:

In this manuscript, the authors attempt to incorporate five consecutive PT linkages to stabilize a short PCR fragment (<50bp) during T5-independent cloning. Overall, the manuscript presents some valuable insights, but several concerns must be addressed before publication:

1. Gibson assembly is a widely used method for DNA assembly, with many commercially available kits that can clone nearly any DNA fragment using T5 exonuclease, Phusion polymerase, and Taq ligase. Compared to this method, what are the main advantages of DAPE? Please elaborate on this in both the Introduction and Discussion sections. Additionally, the authors claim that inserting five consecutive PT linkages (5PT) makes the cloning end resistant to T5 exonuclease, improving the precision of the cloning process. How does this precision compare to Gibson assembly? Please design an experiment to evaluate this.

2. In Fig. 1D, the cloning efficiency significantly decreases after adding the 5PT linkages. Could the authors explain this? Also, please include a statistical analysis for this experiment. Additionally, clarify the full name of "CFU" in the figure legends.

3. Fig. 2A is identical to Fig. 1A. Please either remove it or provide additional information to differentiate it.

4. In Fig. 2B, please provide detailed conditions for the control group.

5. Fig. 3C requires statistical analysis. Please include this in the revised version.

6. On page 12, the sentence "PCR was carried out after direct primer-to-primer annealing, and then 1 μl (5%) of the PCR products was used as inserts without undergoing any DNA cleaning procedure." should be moved before "The 48 bp PCR products were reacted…".

7. Fig. 4B lacks clarity, and there is no direct comparison between the Golden Gate and DAPE groups. Please add a statistical analysis and clarify the meaning of the "Golden Gate ctl group." Ensure that all control groups are consistently labeled across all figures.

8. Fig. 5A is confusing and needs to be presented more clearly. Additionally, since EGFP is merely an example gene, it should not be given undue emphasis in the figure.

9. In Fig. 5B, 5PT actually reduces cloning efficiency in the assembly of longer fragments, as seen in Fig. 1D. Please explain this drawback of 5PT. Also, why are many of the figures missing statistical analyses?

10. Fig. 6A is unclear. Please indicate the position of the 5PT linkages in the figure

11. For comparison to lambda enzyme, if possible, please cite this reference, “Homogeneous and Sensitive Detection of microRNA with Ligase Chain Reaction and Lambda Exonuclease-Assisted Cationic Conjugated Polymer Biosensing”

Reviewer 2:

1. In the Introduction, paragraph 2, the authors state: "SLIC allows any linearizable DNA fragment to be inserted into any position of any vector of choice without incorporating additional DNA sequences, a process known as 'seamless cloning.'" However, this conclusion is limited because it does not explain the specific disadvantages of SLIC, such as the degradation of small DNA fragments due to excessive exonuclease activity. The authors should expand the discussion of the limitations of existing methods like SLIC, Gibson assembly, and Golden Gate, especially in handling small DNA fragments, and cite relevant studies or data to strengthen this point.

2. In the Introduction, paragraph 3, the authors introduce: "We introduce a versatile sequence- and ligation-independent cloning (SLIC) method called 'DNA Assembly with Phosphorothioate (PT) and T5 Exonuclease' (DAPE), which generates precise lengths of 3' overhangs at both ends of linearized DNA." This explanation is somewhat limited, as it does not elaborate on how PT-modified primers and T5 exonuclease achieve this precision. The authors should add details about how T5 exonuclease differs from other commonly used exonucleases, such as T4 DNA polymerase, and how PT-modified primers prevent excessive degradation, to help readers better understand the innovation in DAPE.

3. In the Materials and Methods section, the experimental conditions are not fully detailed, for example: "PCR samples were reacted with 1 U of diluted Lambda Exonuclease...The mixture was incubated for 90 min at 37°C before transformation." This description is too vague and might hinder reproducibility. The authors should specify all critical variables such as the exact concentration of enzymes, reaction times, and temperature controls for each step, ensuring that other researchers can replicate the experiments.

4. In the Results section, most of the experimental validation focuses on fragments smaller than 80 bp, as mentioned: "The 48 bp PCR products were reacted with 0.1 U of T5 exonuclease for 30 minutes at 30°C on a heat block." However, the effects on larger DNA fragments are not tested, which may limit the method's generalizability. The authors should include tests on larger DNA fragments (e.g., 150 bp or longer) and compare the cloning efficiency of these larger fragments with smaller ones to validate DAPE’s effectiveness across a wider range of fragment sizes.

5. In the Results section, statistical analysis is lacking, where the authors state: "Transformation efficiencies were calculated by counting the number of colonies in triplicate experiments." Although the experiments were repeated, no significance testing was performed, making it difficult to determine whether differences are meaningful. The authors should add statistical tests such as t-tests or ANOVA to evaluate the differences in cloning efficiency across conditions and indicate whether these differences are statistically significant.

6. In the Discussion section, the limitations of the DAPE method are not sufficiently addressed, for example: "Therefore, avoiding the use of DNA ligase during DNA assembly would be preferable for increasing the proportion of positive clones." While the advantages of avoiding ligase are highlighted, potential limitations are not discussed, such as the challenges DAPE may face in assembling more complex DNA constructs. The authors should add a discussion of potential limitations of the DAPE method, such as its performance in more complex genome assemblies or when handling larger DNA fragments, to provide a more balanced evaluation of the technique.

7. In the Discussion section, the comparison with other methods is not sufficiently in-depth: "DAPE, as an advanced toolkit for DNA cloning and synthetic biology, may further expedite the construction of more elaborate multi-gene circuitry." However, the actual differences between DAPE and other common methods (such as Gibson and Golden Gate) are not explored in detail. The authors should provide a more detailed comparison with other commonly used methods, particularly how DAPE performs in multi-gene assembly or with larger constructs, to better highlight its advantages in practical applications.

8. In the Figures section, the labeling and annotations are insufficient, as seen in the sentence: "Transformation efficiencies were calculated by counting the number of colonies..." The figures provide the data but lack sufficient annotations, making it difficult for readers to interpret the differences between conditions. The authors should add clearer labels and annotations for all figures, including explanations of each experimental condition and data source, to make the results more accessible to readers.

9. There are insufficient references to support the claims, for instance, when the authors mention: "Gibson-based cloning, which includes Taq ligase, caused relatively high levels of background clones." This statement is made without sufficient citation to support it. The authors should add more citations of recent studies, particularly those related to SLIC, Gibson assembly, and Golden Gate, to support the claims and improve the manuscript's credibility.

10. The experimental design lacks diversity, as only linear DNA fragments were tested: "PCR samples were reacted with 1 U of diluted Lambda Exonuclease and supplemented with 10x buffer..." No tests were conducted on other types of DNA, such as circular DNA or sequences with secondary structures. The authors should include experiments with different types of DNA (e.g., GC-rich sequences, DNA with secondary structures, or circular DNA) to test the robustness of the DAPE method in diverse scenarios.

11. The manuscript lacks discussion of practical application scenarios, for example: "DAPE, as an advanced toolkit for DNA cloning and synthetic biology, may further expedite the construction of more elaborate multi-gene circuitry." However, it does not discuss specific real-world applications, such as in CRISPR/Cas9 systems. The authors should expand the discussion to include the potential of DAPE in real-world research applications, such as CRISPR gene editing or constructing complex gene circuits, to show its practical utility and broaden its appeal.

12. The language is sometimes overly complex and the sentences are too long, as in: "T5 Exonuclease-based cloning was also known as TEDA (T5 exonuclease DNA assembly). To optimize the reaction conditions..." The long sentences may confuse readers. The authors should simplify the language and break up long sentences into shorter, more digestible parts, ensuring that each sentence conveys a single, clear point, to improve readability.

13. The results and discussion are sometimes mixed, for example: "Since the cloning efficiency of multi-piece DNA assembly by TEDA-based SLIC gradually increased with longer complementary single-stranded overhangs..." This includes both result descriptions and discussion, making the structure less clear. The authors should clearly separate the results from the discussion by placing all experimental data and observations in the "Results" section, and reserving the interpretation and analysis for the "Discussion" section, to enhance clarity.

14. The discussion of multi-fragment assembly is not detailed enough, for instance: "The relatively low yield of DAPE for assembling multiple DNA fragments could be improved by extending the 3’ overhang to over 20 bp." While the problem is mentioned, specific solutions are not discussed. The authors should expand the discussion on how to improve the efficiency of multi-fragment assembly. Consider proposing solutions such as adjusting the length of the 3’ overhangs or optimizing primer design, to provide more concrete recommendations for future improvements.

Reviewers' comments:

Reviewer's Responses to Questions

**Comments to the Author**

1. Is the manuscript technically sound, and do the data support the conclusions?

Reviewer #1: Yes

Reviewer #2: Partly

2. Has the statistical analysis been performed appropriately and rigorously? 

Reviewer #1: Yes

Reviewer #2: I Don't Know

3. Have the authors made all data underlying the findings in their manuscript fully available?

Reviewer #1: Yes

Reviewer #2: Yes

4. Is the manuscript presented in an intelligible fashion and written in standard English?

Reviewer #1: Yes

Reviewer #2: Yes

5. Review Comments to the Author

Reviewer #1: In this manuscript, the authors attempt to incorporate five consecutive PT linkages to stabilize a short PCR fragment (<50bp) during T5-independent cloning. Overall, the manuscript presents some valuable insights, but several concerns must be addressed before publication:

1.Gibson assembly is a widely used method for DNA assembly, with many commercially available kits that can clone nearly any DNA fragment using T5 exonuclease, Phusion polymerase, and Taq ligase. Compared to this method, what are the main advantages of DAPE? Please elaborate on this in both the Introduction and Discussion sections. Additionally, the authors claim that inserting five consecutive PT linkages (5PT) makes the cloning end resistant to T5 exonuclease, improving the precision of the cloning process. How does this precision compare to Gibson assembly? Please design an experiment to evaluate this.

2.In Fig. 1D, the cloning efficiency significantly decreases after adding the 5PT linkages. Could the authors explain this? Also, please include a statistical analysis for this experiment. Additionally, clarify the full name of "CFU" in the figure legends.

3.Fig. 2A is identical to Fig. 1A. Please either remove it or provide additional information to differentiate it.

4.In Fig. 2B, please provide detailed conditions for the control group.

5.Fig. 3C requires statistical analysis. Please include this in the revised version.

6.On page 12, the sentence "PCR was carried out after direct primer-to-primer annealing, and then 1 μl (5%) of the PCR products was used as inserts without undergoing any DNA cleaning procedure." should be moved before "The 48 bp PCR products were reacted…".

7.Fig. 4B lacks clarity, and there is no direct comparison between the Golden Gate and DAPE groups. Please add a statistical analysis and clarify the meaning of the "Golden Gate ctl group." Ensure that all control groups are consistently labeled across all figures.

8.Fig. 5A is confusing and needs to be presented more clearly. Additionally, since EGFP is merely an example gene, it should not be given undue emphasis in the figure.

9.In Fig. 5B, 5PT actually reduces cloning efficiency in the assembly of longer fragments, as seen in Fig. 1D. Please explain this drawback of 5PT. Also, why are many of the figures missing statistical analyses?

10.Fig. 6A is unclear. Please indicate the position of the 5PT linkages in the figure

11.For comparison to lambda enzyme, if possible, please cite this reference, “Homogeneous and Sensitive Detection of microRNA with Ligase Chain Reaction and Lambda Exonuclease-Assisted Cationic Conjugated Polymer Biosensing”

Reviewer #2: 1、In the Introduction, paragraph 2, the authors state: "SLIC allows any linearizable DNA fragment to be inserted into any position of any vector of choice without incorporating additional DNA sequences, a process known as 'seamless cloning.'" However, this conclusion is limited because it does not explain the specific disadvantages of SLIC, such as the degradation of small DNA fragments due to excessive exonuclease activity. The authors should expand the discussion of the limitations of existing methods like SLIC, Gibson assembly, and Golden Gate, especially in handling small DNA fragments, and cite relevant studies or data to strengthen this point.

2、In the Introduction, paragraph 3, the authors introduce: "We introduce a versatile sequence- and ligation-independent cloning (SLIC) method called 'DNA Assembly with Phosphorothioate (PT) and T5 Exonuclease' (DAPE), which generates precise lengths of 3' overhangs at both ends of linearized DNA." This explanation is somewhat limited, as it does not elaborate on how PT-modified primers and T5 exonuclease achieve this precision. The authors should add details about how T5 exonuclease differs from other commonly used exonucleases, such as T4 DNA polymerase, and how PT-modified primers prevent excessive degradation, to help readers better understand the innovation in DAPE.

3、In the Materials and Methods section, the experimental conditions are not fully detailed, for example: "PCR samples were reacted with 1 U of diluted Lambda Exonuclease...The mixture was incubated for 90 min at 37°C before transformation." This description is too vague and might hinder reproducibility. The authors should specify all critical variables such as the exact concentration of enzymes, reaction times, and temperature controls for each step, ensuring that other researchers can replicate the experiments.

4、In the Results section, most of the experimental validation focuses on fragments smaller than 80 bp, as mentioned: "The 48 bp PCR products were reacted with 0.1 U of T5 exonuclease for 30 minutes at 30°C on a heat block." However, the effects on larger DNA fragments are not tested, which may limit the method's generalizability. The authors should include tests on larger DNA fragments (e.g., 150 bp or longer) and compare the cloning efficiency of these larger fragments with smaller ones to validate DAPE’s effectiveness across a wider range of fragment sizes.

5、In the Results section, statistical analysis is lacking, where the authors state: "Transformation efficiencies were calculated by counting the number of colonies in triplicate experiments." Although the experiments were repeated, no significance testing was performed, making it difficult to determine whether differences are meaningful. The authors should add statistical tests such as t-tests or ANOVA to evaluate the differences in cloning efficiency across conditions and indicate whether these differences are statistically significant.

6、In the Discussion section, the limitations of the DAPE method are not sufficiently addressed, for example: "Therefore, avoiding the use of DNA ligase during DNA assembly would be preferable for increasing the proportion of positive clones." While the advantages of avoiding ligase are highlighted, potential limitations are not discussed, such as the challenges DAPE may face in assembling more complex DNA constructs. The authors should add a discussion of potential limitations of the DAPE method, such as its performance in more complex genome assemblies or when handling larger DNA fragments, to provide a more balanced evaluation of the technique.

7、In the Discussion section, the comparison with other methods is not sufficiently in-depth: "DAPE, as an advanced toolkit for DNA cloning and synthetic biology, may further expedite the construction of more elaborate multi-gene circuitry." However, the actual differences between DAPE and other common methods (such as Gibson and Golden Gate) are not explored in detail. The authors should provide a more detailed comparison with other commonly used methods, particularly how DAPE performs in multi-gene assembly or with larger constructs, to better highlight its advantages in practical applications.

8、In the Figures section, the labeling and annotations are insufficient, as seen in the sentence: "Transformation efficiencies were calculated by counting the number of colonies..." The figures provide the data but lack sufficient annotations, making it difficult for readers to interpret the differences between conditions. The authors should add clearer labels and annotations for all figures, including explanations of each experimental condition and data source, to make the results more accessible to readers.

9、There are insufficient references to support the claims, for instance, when the authors mention: "Gibson-based cloning, which includes Taq ligase, caused relatively high levels of background clones." This statement is made without sufficient citation to support it. The authors should add more citations of recent studies, particularly those related to SLIC, Gibson assembly, and Golden Gate, to support the claims and improve the manuscript's credibility.

10、The experimental design lacks diversity, as only linear DNA fragments were tested: "PCR samples were reacted with 1 U of diluted Lambda Exonuclease and supplemented with 10x buffer..." No tests were conducted on other types of DNA, such as circular DNA or sequences with secondary structures. The authors should include experiments with different types of DNA (e.g., GC-rich sequences, DNA with secondary structures, or circular DNA) to test the robustness of the DAPE method in diverse scenarios.

11、The manuscript lacks discussion of practical application scenarios, for example: "DAPE, as an advanced toolkit for DNA cloning and synthetic biology, may further expedite the construction of more elaborate multi-gene circuitry." However, it does not discuss specific real-world applications, such as in CRISPR/Cas9 systems. The authors should expand the discussion to include the potential of DAPE in real-world research applications, such as CRISPR gene editing or constructing complex gene circuits, to show its practical utility and broaden its appeal.

12、The language is sometimes overly complex and the sentences are too long, as in: "T5 Exonuclease-based cloning was also known as TEDA (T5 exonuclease DNA assembly). To optimize the reaction conditions..." The long sentences may confuse readers. The authors should simplify the language and break up long sentences into shorter, more digestible parts, ensuring that each sentence conveys a single, clear point, to improve readability.

13、The results and discussion are sometimes mixed, for example: "Since the cloning efficiency of multi-piece DNA assembly by TEDA-based SLIC gradually increased with longer complementary single-stranded overhangs..." This includes both result descriptions and discussion, making the structure less clear. The authors should clearly separate the results from the discussion by placing all experimental data and observations in the "Results" section, and reserving the interpretation and analysis for the "Discussion" section, to enhance clarity.

14、The discussion of multi-fragment assembly is not detailed enough, for instance: "The relatively low yield of DAPE for assembling multiple DNA fragments could be improved by extending the 3’ overhang to over 20 bp." While the problem is mentioned, specific solutions are not discussed. The authors should expand the discussion on how to improve the efficiency of multi-fragment assembly. Consider proposing solutions such as adjusting the length of the 3’ overhangs or optimizing primer design, to provide more concrete recommendations for future improvements.

6. PLOS authors have the option to publish the peer review history of their article (what does this mean?). If published, this will include your full peer review and any attached files.

Reviewer #1: No

Reviewer #2: No

---

## [Author Response · Author response to Decision Letter 0]

4 Jan 2025

Reviewer #1: In this manuscript, the authors attempt to incorporate five consecutive PT linkages to stabilize a short PCR fragment (<50bp) during T5-independent cloning. Overall, the manuscript presents some valuable insights, but several concerns must be addressed before publication:

1. Gibson assembly is a widely used method for DNA assembly, with many commercially available kits that can clone nearly any DNA fragment using T5 exonuclease, Phusion polymerase, and Taq ligase. Compared to this method, what are the main advantages of DAPE? Please elaborate on this in both the Introduction and Discussion sections. Additionally, the authors claim that inserting five consecutive PT linkages (5PT) makes the cloning end resistant to T5 exonuclease, improving the precision of the cloning process. How does this precision compare to Gibson assembly? Please design an experiment to evaluate this. 

-> Following the reviewer's comments, we conducted experiments to compare DAPE and Gibson assembly, as shown in Figure 4. The results demonstrated that incorporating 5PT into the primer significantly enhanced the cloning efficiency of small DNA fragments (48 bp) in both DAPE and Gibson assembly. Thus, 5PT-dependent DAPE could be applied to Gibson assembly to overcome its limitations in subcloning small DNA fragments.

2. In Fig. 1D, the cloning efficiency significantly decreases after adding the 5PT linkages. Could the authors explain this? Also, please include a statistical analysis for this experiment. Additionally, clarify the full name of "CFU" in the figure legends. 

-> In response to the reviewer's comments, we performed t-tests and added p-values to all graphs after counting individual colonies. Additionally, we clarified abbreviations, including CFU, in the figure legends. However, as we observed a tendency for reduced cloning efficiency when testing relatively long DNA fragments with 5PT for no apparent rational reason, we recommend that 5PT be used primarily for cloning short DNA fragments or preventing secondary structure formation at the ends of DNA fragments. We addressed the issues raised by the reviewer in the Discussion section. The detailed content is provided below:

“Additionally, it is important to note that the 5PT modification was not advantageous for relatively long DNA fragments, as shown in Figure 1. Therefore, we suggest that 5PT should be applied to DNA fragments shorter than 80 bp, as demonstrated in Figure 6.”

3. Fig. 2A is identical to Fig. 1A. Please either remove it or provide additional information to differentiate it. 

-> Following reviewer’s advices, we removed Fig. 2A. 

4. In Fig. 2B, please provide detailed conditions for the control group. 

-> We appreciate the reviewer for pointing out the ambiguities. We have added a more detailed explanation of the control in both the main text and the Figure 2 legend.

“Main text; As a negative control, we used an equal amount of linear DNA mixture without T5 exonuclease (-T5 Exo).”

“Figure 2 legend; A linear DNA mixture without T5 exonuclease (-T5 Exo) was used as a negative control.”

5. Fig. 3C requires statistical analysis. Please include this in the revised version. 

-> Thank you for the advice. We performed t-tests and added p-values to all graphs.

6. On page 12, the sentence "PCR was carried out after direct primer-to-primer annealing, and then 1 μl (5%) of the PCR products was used as inserts without undergoing any DNA cleaning procedure." should be moved before "The 48 bp PCR products were reacted…". 

-> Following the reviewer's advice, we moved the sentence to precede the indicated one.

7. Fig. 4B lacks clarity, and there is no direct comparison between the Golden Gate and DAPE groups. Please add a statistical analysis and clarify the meaning of the "Golden Gate ctl group." Ensure that all control groups are consistently labeled across all figures. 

-> Thank you to the reviewer for pointing out the ambiguities. We have added an explanation of the control, as shown below, in the Figure 5 legend (Figure 4 has been renumbered as Figure 5). We also added the p-value after performing the t-test.

“A linear vector without gRNA was used as a negative control (labeled as "linear vector").”

8. Fig. 5A is confusing and needs to be presented more clearly. Additionally, since EGFP is merely an example gene, it should not be given undue emphasis in the figure.

 -> Following the reviewer's advice, we changed the labeling from EGFP to PCR template in Figure 6A and provided a clear explanation in the legend (Figure 5 has been renumbered as Figure 6.).

9. In Fig. 5B, 5PT actually reduces cloning efficiency in the assembly of longer fragments, as seen in Fig. 1D. Please explain this drawback of 5PT. Also, why are many of the figures missing statistical analyses? 

->Following the reviewer’s valuable advice, we added the p-value to the graph and provided an explanation regarding the limitation of 5PT in the Discussion section, as follows (The Figure 5 has been renumbered as Figure 6.)

“Additionally, it is important to note that the 5PT modification was not advantageous for relatively long DNA fragments, as shown in Figure 1. Therefore, we suggest that 5PT should be applied to DNA fragments shorter than 80 bp, as demonstrated in Figure 6.”

10. Fig. 6A is unclear. Please indicate the position of the 5PT linkages in the figure

-> Following reviewer’s request, we added asterisks on the primers as the position of 5PT. We also indicated the meaning of the asterisks in Figure 8 legend such as “Asterisks on the primers represent 5PT linkages.” (The Figure 6 becomes Figure 8.)

11. For comparison to lambda enzyme, if possible, please cite this reference, “Homogeneous and Sensitive Detection of microRNA with Ligase Chain Reaction and Lambda Exonuclease-Assisted Cationic Conjugated Polymer Biosensing”

-> We appreciate the valuable reference paper provided and have cited the recommended paper.

Reviewer #2: 

1、 In the Introduction, paragraph 2, the authors state: "SLIC allows any linearizable DNA fragment to be inserted into any position of any vector of choice without incorporating additional DNA sequences, a process known as 'seamless cloning.'" However, this conclusion is limited because it does not explain the specific disadvantages of SLIC, such as the degradation of small DNA fragments due to excessive exonuclease activity. The authors should expand the discussion of the limitations of existing methods like SLIC, Gibson assembly, and Golden Gate, especially in handling small DNA fragments, and cite relevant studies or data to strengthen this point. 

-> Following the valuable advice of the reviewer, we conducted experiments to compare Gibson assembly and DAPE, as shown in Figure 4. Additionally, we expanded our explanation of the possible and practical limitations of current SLIC methods and highlighted the additional potential value of DAPE cloning in addressing these challenges in the Discussion section.

2、 In the Introduction, paragraph 3, the authors introduce: "We introduce a versatile sequence- and ligation-independent cloning (SLIC) method called 'DNA Assembly with Phosphorothioate (PT) and T5 Exonuclease' (DAPE), which generates precise lengths of 3' overhangs at both ends of linearized DNA." This explanation is somewhat limited, as it does not elaborate on how PT-modified primers and T5 exonuclease achieve this precision. The authors should add details about how T5 exonuclease differs from other commonly used exonucleases, such as T4 DNA polymerase, and how PT-modified primers prevent excessive degradation, to help readers better understand the innovation in DAPE. 

->Thank you for pointing out the issues that should be discussed in more detail. The sentence "We introduce a versatile sequence- and ligation-independent cloning (SLIC) method called 'DNA Assembly with Phosphorothioate (PT) and T5 Exonuclease' (DAPE), which generates precise lengths of 3' overhangs at both ends of linearized DNA" is in the Abstract not in Introduction section. However, we rewrote the Introduction section, adding a more detailed explanation of the rationale for choosing the T5 exonuclease instead of using other enzymes. The detailed replaced content is provided below emphasized with read characters:

“(original) In this study, we introduce an alternative SLIC method, designated DAPE, which utilizes primers of PT internucleotide linkages to precisely generate the desired length of 3' single-stranded ends in PCR products. The PT internucleotide linkages have been reported to render DNA resistant to nuclease activities without affecting nucleotide base pairing (11, 12). More interestingly, precisely measured 3’ overhangs of 10 bp using PT linkages, by virtue of lambda exonuclease, facilitated large DNA integration into a specific DNA breakpoint generated by CRISPR/Cas9 (12). We also found that incorporating five consecutive PT linkages in the middle of primers makes the PTs near the ends of the PCR product highly resistant to T5 exonuclease activity. This allows for the precise production of predetermined lengths of 3’ recessed ends by T5 exonuclease. This approach offers a highly reliable and versatile method for the generation of recombinant DNA, regardless of the length of DNA fragments. Additionally, we further optimized the T5 exonuclease-based SLIC procedure, which performed superiorly to other exonuclease-based SLIC methods for both single insert and multi-piece cloning into a vector. DNA fragments as short as 50 bp, generated by PCR without undergoing a cleaning procedure, can be assembled in a single reaction with high precision and remarkable efficiency.”

-> “(revised) In this study, we introduce an alternative SLIC method, designated DAPE, which utilizes primers of PT internucleotide linkages to precisely generate the desired length of 3' single-stranded ends in PCR products. The PT internucleotide linkages have been reported to render DNA resistant to nuclease activities without affecting nucleotide base pairing (15, 16). More interestingly, precisely measured 3’ overhangs of 10 bp using PT linkages, by virtue of lambda exonuclease, facilitated large DNA integration into a specific DNA breakpoint generated by CRISPR/Cas9 (16). We also found that incorporating five consecutive PT (5PT) linkages in the middle of primers makes the PTs near the ends of the PCR product highly resistant to T5 exonuclease activity. Because 3’ overhangs cannot be produced by exonucleases with 3’ to 5’ gnawing activity, such as T4 DNA polymerase and the Klenow fragment, which leave 5’ overhangs, we cannot apply them to DAPE. Since the T5 enzyme has been reported to have stronger exonuclease activity than lambda exonuclease, we ultimately chose the T5 enzyme for DAPE. This allows for the precise production of predetermined lengths of 3’ recessed ends by T5 exonuclease. This approach offers a highly reliable and versatile method for the generation of recombinant DNA, regardless of the length of DNA fragments. Additionally, we further optimized the T5 exonuclease-based SLIC procedure, which performed superiorly to other exonuclease-based SLIC methods for both single insert and multi-piece cloning into a vector. DNA fragments as short as 50 bp, generated by PCR without undergoing a cleaning procedure, can be assembled in a single reaction with high precision and remarkable efficiency.”

3、 In the Materials and Methods section, the experimental conditions are not fully detailed, for example: "PCR samples were reacted with 1 U of diluted Lambda Exonuclease...The mixture was incubated for 90 min at 37°C before transformation." This description is too vague and might hinder reproducibility. The authors should specify all critical variables such as the exact concentration of enzymes, reaction times, and temperature controls for each step, ensuring that other researchers can replicate the experiments. 

-> Thank you for the advice. For the convenience of potential readers, we moved the detailed, step-by-step DAPE protocol from the Supplementary Information into the Experimental Procedures.

4、 In the Results section, most of the experimental validation focuses on fragments smaller than 80 bp, as mentioned: "The 48 bp PCR products were reacted with 0.1 U of T5 exonuclease for 30 minutes at 30°C on a heat block." However, the effects on larger DNA fragments are not tested, which may limit the method's generalizability. The authors should include tests on larger DNA fragments (e.g., 150 bp or longer) and compare the cloning efficiency of these larger fragments with smaller ones to validate DAPE’s effectiveness across a wider range of fragment sizes. 

-> Thank you for the advice. According to the reviewer’s comment, we conducted DAPE with fragments ranging from 50 bp to 170 bp, increasing the length by 30 base pairs each, as shown in Figure 6. The results suggested that if the insert is longer than 80 bp, 5PT modification of the primers is not required for subcloning.

5、 In the Results section, statistical analysis is lacking, where the authors state: "Transformation efficiencies were calculated by counting the number of colonies in triplicate experiments." Although the experiments were repeated, no significance testing was performed, making it difficult to determine whether differences are meaningful. The authors should add statistical tests such as t-tests or ANOVA to evaluate the differences in cloning efficiency across conditions and indicate whether these differences are statistically significant. 

-> We appreciate the reviewer pointing out the important issues we unwittingly ignored, such as the t-test on the figures. We conducted a t-test on all the graphical data and displayed the statistical values on the graph

6、In the Discussion section, the limitations of the DAPE method are not sufficiently addressed, for example: "Therefore, avoiding the use of DNA ligase during DNA assembly would be preferable for increasing the proportion of positive clones." While the advantages of avoiding ligase are highlighted, potential limitations are not discussed, such as the challenges DAPE may face in assembling more complex DNA constructs. The authors should add a discussion of potential limitations of the DAPE method, such as its performance in more complex genome assemblies or when handling larger DNA fragments, to provide a more balanced evaluation of the technique. 

-> The potential limitations, as well as the pros and cons of DAPE compared to Gibson assembly, are discussed in the Discussion section. Thanks for the advice. The detailed content is provided below:

“In comparison to Gibson assembly, the potential advantages and disadvantages of DAPE are as follows: First, by omitting DNA ligases, researchers can suppress the emergence of self-ligated clones, albeit at the cost of reducing the overall number of colonies, as shown in Fig. 5 and Fig. 7. Previous reports indicate that while Gibson-based cloning, which includes Taq ligase, caused relatively high levels of background clones, other simplified Gibson techniques, such as Hot Fusion and TEDA, which are free of DNA ligase, produced fewer self-ligated clones without affecting cloning efficiency (8, 10, 11). Second, we can achieve optimal results by using 5PT-incorporated primers in combination with Gibson assembly to subclone a DNA that contains sequences prone to forming secondary structures at both ends (Fig. 7). There are two rational explanations for the improved cloning efficiency of 5PT with Gibson assembly. First, 5PT may protect sequences located more than 15 bp from the 5' end, preventing the remaining 3' single-strand from forming a hairpin structure. Second, while DAPE was performed at 30°C, Gibson assembly was conducted at 50°C, a relatively high temperature that likely helps prevent secondary structure formation. Considering the results in Fig. 4, DAPE is a preferable choice for constructing plasmids with small DNA fragments. However, under more complex conditions, such as concerns about secondary structure formation at the ends of the DNA fragment, Gibson assembly with 5PT-modified primers can enhance overall cloning efficiency.”

6、 In the Discussion section, the comparison with other methods is not sufficiently in-depth: "DAPE, as an adva

---

## [Decision Letter · Decision Letter 1]

8 Jan 2025

DAPE Cloning with Modified Primers for Producing Designated Lengths of 3’ Single-Stranded Ends in PCR Products

PONE-D-24-43146R1

Dear Dr. Ro,

We’re pleased to inform you that your manuscript has been judged scientifically suitable for publication and will be formally accepted for publication once it meets all outstanding technical requirements.

Kind regards,

Zheng Yuan

Academic Editor

PLOS ONE

Additional Editor Comments (optional):

Reviewers' comments:

Reviewer's Responses to Questions

**Comments to the Author**

1. If the authors have adequately addressed your comments raised in a previous round of review and you feel that this manuscript is now acceptable for publication, you may indicate that here to bypass the “Comments to the Author” section, enter your conflict of interest statement in the “Confidential to Editor” section, and submit your "Accept" recommendation.

Reviewer #1: All comments have been addressed

Reviewer #2: All comments have been addressed

2. Is the manuscript technically sound, and do the data support the conclusions?

Reviewer #1: Yes

Reviewer #2: Yes

3. Has the statistical analysis been performed appropriately and rigorously? 

Reviewer #1: Yes

Reviewer #2: I Don't Know

4. Have the authors made all data underlying the findings in their manuscript fully available?

Reviewer #1: Yes

Reviewer #2: Yes

5. Is the manuscript presented in an intelligible fashion and written in standard English?

Reviewer #1: Yes

Reviewer #2: Yes

6. Review Comments to the Author

Reviewer #1: (No Response)

Reviewer #2: (No Response)

7. PLOS authors have the option to publish the peer review history of their article (what does this mean?). If published, this will include your full peer review and any attached files.

Reviewer #1: No

Reviewer #2: No

---

## [Editor Report · Acceptance letter]

15 Jan 2025

PONE-D-24-43146R1 

PLOS ONE

Dear Dr. Ro, 

I'm pleased to inform you that your manuscript has been deemed suitable for publication in PLOS ONE. Congratulations! Your manuscript is now being handed over to our production team.

Kind regards, 

on behalf of

Dr. Zheng Yuan 

Academic Editor

PLOS ONE